# Natriuretic Peptide Signaling in Uterine Biology and Preeclampsia

**DOI:** 10.3390/ijms241512309

**Published:** 2023-08-01

**Authors:** Qingyu Wu

**Affiliations:** Cyrus Tang Hematology Center, Collaborative Innovation Center of Hematology, State Key Laboratory of Radiation Medicine and Prevention, Soochow University, Suzhou 215123, China; wuqy@suda.edu.cn

**Keywords:** ANP, corin, endometrial decidualization, natriuretic peptides, preeclampsia, spiral artery remodeling

## Abstract

Endometrial decidualization is a uterine process essential for spiral artery remodeling, embryo implantation, and trophoblast invasion. Defects in endometrial decidualization and spiral artery remodeling are important contributing factors in preeclampsia, a major disorder in pregnancy. Atrial natriuretic peptide (ANP) is a cardiac hormone that regulates blood volume and pressure. ANP is also generated in non-cardiac tissues, such as the uterus and placenta. In recent human genome-wide association studies, multiple loci with genes involved in natriuretic peptide signaling are associated with gestational hypertension and preeclampsia. In cellular experiments and mouse models, uterine ANP has been shown to stimulate endometrial decidualization, increase TNF-related apoptosis-inducing ligand expression and secretion, and enhance apoptosis in arterial smooth muscle cells and endothelial cells. In placental trophoblasts, ANP stimulates adenosine 5′-monophosphate-activated protein kinase and the mammalian target of rapamycin complex 1 signaling, leading to autophagy inhibition and protein kinase N3 upregulation, thereby increasing trophoblast invasiveness. ANP deficiency impairs endometrial decidualization and spiral artery remodeling, causing a preeclampsia-like phenotype in mice. These findings indicate the importance of natriuretic peptide signaling in pregnancy. This review discusses the role of ANP in uterine biology and potential implications of impaired ANP signaling in preeclampsia.

## 1. Introduction

Pregnancy is associated with major hormonal changes, including increased production and release of chorionic gonadotropin from trophoblasts and progesterone from the corpus luteum in the ovary. These hormones induce profound cellular and functional changes in the uterine tissue, e.g., endometrial decidualization and spiral artery remodeling. Such changes are essential for embryo implantation and an adequate uteroplacental blood flow to support the developing placenta and fetus.

Preeclampsia is a disorder characterized by the new onset of high blood pressure after ~20 weeks of pregnancy [1,2]. It occurs in ~4–8% of all pregnancies and often leads to damages in major organs, including the kidney, liver, and placenta [3]. The cause of preeclampsia is not fully understood. One of the common pathological findings in the preeclamptic uterus is incompletely remodeled spiral arteries, which limits blood flow to the placenta [4,5]. Placental ischemia and oxidative stress in turn lead to the release of detrimental placental factors into the maternal circulation, causing inflammatory responses, endothelial dysfunction, and systemic complications [4,6]. It has been shown that aberrant expression and/or function of angiogenic factors, e.g., vascular endothelial growth factor (VEGF) and its receptors, e.g., soluble VEGF receptor 1, also called soluble fms-like tyrosine kinase-1 (sFlt1), play a major role in the pathogenesis of preeclampsia [1,7].

Poor endometrial decidualization, also called decidualization resistance, is another pathological characteristic in preeclampsia [8,9,10]. Decreased circulating levels of insulin-like growth factor-binding protein-1 (IGFBP-1), an endometrial decidualization marker, in preeclamptic women are well documented [9,11]. These findings indicate that impaired molecular mechanisms controlling uterine endometrial decidualization and tissue remodeling are potential contributing factors in preeclampsia.

Atrial natriuretic peptide (ANP) is a versatile hormone produced largely in cardiomyocytes [12,13,14]. The main function of ANP is to control electrolyte homeostasis, body fluid balance, and blood pressure. Studies in cultured cells and mouse models indicate that ANP also acts in the pregnant uterus to stimulate endometrial decidualization, spiral artery remodeling, and trophoblast invasion [15,16,17]. Recent large-scale genetic studies in humans have also implicated impaired ANP generation and signaling in gestational hypertension and preeclampsia [18,19]. This review provides a brief overview of ANP biosynthesis and function in cardiovascular biology and metabolism. The main discussion will be on the role of ANP and its activating protease corin in uterine tissue remodeling and the potential implications of ANP and corin deficiency in gestational hypertension and preeclampsia.

## 2. Overview of ANP Biosynthesis, Signaling, and Function

### 2.1. Evolutionary Origin

ANP is a member of the mammalian natriuretic peptide family, which has two additional structurally related peptides, i.e., B-type or brain natriuretic peptide (BNP) and C-type natriuretic peptide (CNP). The *NPPA* gene, encoding the ANP precursor, and the *NPPB* gene, encoding the BNP precursor, are in tandem on the reverse strain of human chromosome 1 between nucleotide positions 11,845,709–11,848,345 and 11,857,464–11,858,945, respectively (Ensembl genome browser, version 109). The *NPPC* gene, encoding the CNP precursor, is on the reverse strain of human chromosome 2 between nucleotide positions 231,921,809 and 231,926,396 (Ensembl genome browser, version 109). These three genes have similar genomic structures, with three exons in each. The similarity suggests that these genes may have evolved from an ancestral gene via gene duplication.

The evolutionary origin of the natriuretic peptide genes can be traced back to early vertebrates such as primitive fish species, in which electrolyte homeostasis is critical for survival in salty waters [20,21]. Based on genomic analyses, natriuretic peptide homologues are present in virtually all vertebrates, ranging from fish, amphibians, reptiles, and birds, to mammals (Ensembl genome browser, version 109). The evolutionary history and conservation underscore the physiological significance of these peptides in the adaptation of vertebrates in aquatic and terrestrial environments [22,23,24].

### 2.2. Proteolytic Processing

The human *NPPA* gene encodes a polypeptide, i.e., prepro-ANP, consisting of a 25-amino-acid signal peptide, a 98-amino-acid propeptide, and the mature ANP of 28 amino acids [12]. The signal peptide, critical for protein secretion, is cleaved by the signal peptidase in the endoplasmic reticulum (ER). The resulting pro-ANP is stored in the dense granules of cardiomyocytes [25]. Upon secretion, pro-ANP is converted to ANP on the cell surface by corin, a type II transmembrane serine protease [26,27,28] (Figure 1). Like ANP, corin is expressed mostly in cardiomyocytes [29,30]. Low levels of corin expression have also been detected in non-cardiac tissues, including the kidney, skin, uterus, and developing bones [30,31,32,33,34,35,36,37]. In *Corin* knockout (KO) mice, the conversion of pro-ANP to ANP in the heart is eliminated, indicating that corin is essential for ANP activation [38]. In the kidney, pro-ANP can be cleaved at an alternative site, resulting in a peptide, called urodilatin, which has four additional amino acids to the N-terminus of ANP [39]. The enzyme responsible for the alternative cleavage probably is a metalloproteinase based on the cleavage site, but its identity remains undefined.

Although the natriuretic peptides share a high degree of similarities in gene structure and protein sequence, the processing mechanisms for these peptides vary. For pro-BNP, both corin and furin, a widely expressed proprotein convertase, exhibit the processing activity in cultured cells [31,40,41]. In cardiomyocytes, however, furin appears the main protease for pro-BNP processing [42]. Furin is also responsible for pro-CNP processing [43,44]. The involvement of different proteolytic enzymes in pro-ANP, pro-BNP, and pro-CNP processing may reflect the divergence in cellular distribution and biological function of these peptides. In general, ANP and BNP have overlapping activities in regulating electrolyte and cardiovascular homeostasis, whereas CNP is involved in cell growth, differentiation, and other cellular events in diverse tissues, including the heart [45,46]. A recent study in mice also indicates the role of CNP in regulating adiposity and thermogenesis [47]. In the following sections, the discussion will focus mostly on molecules related to ANP signaling and function.

### 2.3. ANP Receptors and Signaling

There are two types of ANP receptors: the natriuretic peptide receptor A (NPR-A), also known as the guanylyl cyclase-A receptor (GC-A), and the natriuretic peptide receptor-C (NPR-C) [46,48]. NPR-A is the primary receptor that mediates the biological function of ANP, whereas NPR-C is the clearance receptor that removes ANP from the circulation via endocytosis and intracellular degradation.

NPR-A is present on the surface of many cell types, including vascular endothelial cells, smooth muscle cells, and renal epithelial cells [46,48]. Upon ANP binding, NPR-A undergoes a conformational change, which activates the guanylyl cyclase in its cytoplasmic segment, leading to the conversion of guanosine triphosphate to cyclic guanosine monophosphate (cGMP). In cells, cGMP serves as a second messenger that activates various downstream proteins, particularly protein kinase G (PKG). PKG activation increases the phosphorylation of its effectors, which alter diverse cellular functions, such as gene expression, protein secretion, ion channel activity, and cell contractility, resulting in many physiological changes in targeted tissues [46,48].

NPR-C is a clearance receptor for ANP, BNP, and CNP [46,48]. It is present in most tissues, including the heart, blood vessel, bone, kidney, and adrenal gland. Unlike NPR-A, NPR-C does not have a guanylyl cyclase module in the cytoplasmic segment, and thus does not generate intracellular cGMP. The main function of NPR-C is to remove natriuretic peptides from the circulation by internalization and the subsequent degradation. However, there is evidence that NPR-C can produce alternative intracellular signals via a mechanism linked to inhibitory G proteins and phospholipase C activation. This signaling mechanism may play a role in cardiac fibroblast growth and vascular smooth muscle cell relaxation [49].

### 2.4. ANP Functions in Sodium Homeostasis, Cardiovascular Biology, and Lipid Metabolism

#### 2.4.1. Sodium Homeostasis

ANP is a key natriuretic hormone. In the kidney, ANP enhances the glomerular filtration rate and hence the excretion of sodium and water [46,50]. In the renal tubules, ANP inhibits sodium reabsorption by suppressing type IIa Na/Pi cotransporter, the Na^+^-K^+^-ATPase pump, and Na^+^/H^+^ exchanger in the proximal tubule and sodium channels, e.g., the epithelial sodium channel (ENaC), in the medullary ducts [50]. In the adrenal glands, ANP inhibits the release of aldosterone, a central hormone in promoting sodium conservation [51]. As shown in recent studies, locally generated ANP by corin in the skin eccrine sweat glands and the intestine also antagonizes aldosterone action and promotes sodium excretion in the skin and intestine, respectively [33,52].

#### 2.4.2. Vasodilation

ANP promotes vasodilation by decreasing cytosolic Ca^2+^ concentration and relaxing myosin light chains in vascular smooth muscle cells and stimulating the release of endothelial nitric oxide, a potent vasodilator [53]. Additionally, ANP inhibits the renin release from the renal juxtaglomerular cells and lowers sympathetic nervous system activity, thereby decreasing vasoconstriction and vascular resistance. These ANP functions are important in preventing high blood pressure. Consistently, variants associated with the *NPPA* and *NPPB* genes have been identified as a significant factor in influencing blood pressure levels in human populations [54].

#### 2.4.3. Cardiac Remodeling

In addition to its systemic effects on blood volume and pressure, ANP acts locally in the heart to regulate tissue remodeling, preventing cardiac hypertrophy in response to pathophysiological stress [55,56]. In cardiomyocytes, for example, ANP-mediated cGMP generation activates cGMP-dependent protein kinase I (cGKI) and downstream signaling, to inhibit hypertrophic responses [57,58]. The increased cGMP levels also modulate cardiac contractility in the early phases of cardiac hypertrophy by stimulating β1-adrenergic receptor/cAMP signaling and lowering β2-adrenergic receptor/cAMP signaling in a phosphodiesterase-dependent mechanism [59]. Additionally, ANP inhibits tissue fibrosis in response to cardiac injury by inhibiting TGFβ signaling and reducing cardiac aldosterone and endothelin-1 expression [60]. These ANP actions are important in the maintenance of normal cardiac morphology and function.

#### 2.4.4. Lipid Metabolism and Adipose Tissue Phenotype

Increasing evidence indicates a critical role for ANP in regulating lipid metabolism and adipose tissue phenotype [61,62]. In skeletal muscles, for example, ANP-induced cGMP levels increase PPARγ coactivator-1α and oxidative phosphorylation gene expression, lipid oxidation, and oxygen consumption rate [63]. In adipocytes, ANP stimulates lipolysis by activating the hormone-sensitive lipase in a cGMP and cGKI-mediated signaling mechanism [64]. In humans, ANP treatment enhances lipid oxidation and hydrolysis in adipose tissue [65]. Additionally, ANP promotes the browning program in adipocytes, as indicated by increased uncoupling protein 1 (UCP1) and mitochondrial gene expression, via PKG-p38 mitogen-activated protein kinase (MAPK) signaling [66]. Consistently, *Nppa* and *Corin* KO mice exhibit impaired adipose tissue browning and poor thermogenic response upon cold exposure [67,68].

## 3. Expression of ANP and Corin in the Uterus

The uterus undergoes substantial structural and functional changes during the menstrual cycle and pregnancy [7]. Decidualization is a process in which the endometrial stromal cells are transformed into specialized decidual cells in preparation for receiving a fertilized embryo [7,69]. The process is initiated after ovulation. In humans, embryo implantation is not a prerequisite for endometrial decidualization, as the process also occurs in normal menstrual cycles [7,69] (Figure 2). Progesterone signaling is a central mechanism in promoting endometrial decidualization [7,69]. Following embryo implantation, endometrial decidualization is accelerated, leading to major structural changes in endometrial arteries, such as smooth muscle cell hyperplasia and disarray, endothelial vacuolation, and arterial lumen enlargement [70]. Subsequent placental trophoblast invasion further enhances endometrial decidualization (Figure 2). As described below, ANP and related molecules are expressed in the non-pregnant and pregnant uteruses, suggesting that ANP signaling may play a role in the regulation of uterine biology and function.

### 3.1. Uterine NPPA and NPR1 Expression

ANP, encoded by the *NPPA* gene, is in the circulation. Detection of ANP in non-cardiac tissues may not allow the distinguishing of the cardiac origin from the non-cardiac origin of the detected antigen. Gene expression is a more specific way to analyze the potential expression of ANP protein in non-cardiac tissues. In humans, *NPPA* mRNA has been detected in the non-pregnant uterus [71] and in myometrial cells, decidual cells, and trophoblasts of the pregnant uterus [72,73]. Similar uterine *Nppa* mRNA expression has been found in rodents [16,74,75]. Compared to the cardiac expression, levels of uterine *NPPA* mRNA expression are low. In agreement with these findings, low levels of uterine *NPPA* expression were found by single-cell RNA sequencing in human endometrial stromal cells and glandular and luminal cells (www.proteinatlas.org (accessed on 25 May 2023)). Similarly, *NPR1* (encoding NPR-A) expression in endometrial and myometrial cells has also been found in humans and rodents [72,76,77]. Human uterine *NPR1* expression is mostly in endometrial stromal cells and glandular and luminal cells (www.proteinatlas.org (accessed on 25 May 2023)). These findings suggest that ANP signaling may have a specific function in the regulation of cellular events in the uterus.

### 3.2. Uterine Corin Expression

The *NPPA* gene encodes the ANP precursor. If uterine ANP has a function, uterine corin expression is required to generate ANP. Indeed, corin expression has been reported in human uterine tissues, including the late secretory endometrium and the first trimester decidua [16,34]. Similar *Corin* upregulation was found in the mouse pregnant uterus [16]. As in ANP expression, uterine corin level is lower than that of cardiac corin. In single-cell transcriptomics analysis, *CORIN* mRNA was detected exclusively in a subset of decidual stromal lineage cells [78]. The expression was markedly increased when the cells were treated with cyclic adenosine monophosphate (cAMP) and progesterone.

Unlike those in humans, endometrial decidualization, uterine artery remodeling, and trophoblast invasion are less extensive in many non-primate mammalian species, particularly in rodents [79,80]. Compared to the gene expression profiles in other placental animals, endometrial *CORIN* expression appears enriched in humans, indicating that uterine corin may contribute to human-specific traits in pregnancy, e.g., deep placental invasion and extensive vascular remodeling [78].

In cardiomyocytes, *NPPA* and *CORIN* expression is controlled by the transcription factor GATA-4 [81,82], which is mostly in the heart but not the uterus [83]. In cultured decidual stromal cells, downregulation of the progesterone receptor and GATA-2 reduced *CORIN* promoter activity [78]. In another study, deletion of Krüppel-like factor (KLF) binding sites in the *CORIN* promoter also suppressed the promoter activity in human endometrial cells [84]. Further studies indicate that KLF17 is a critical transcription factor in uterine *CORIN* expression. Disruption of the *KLF17* gene abolished *CORIN* expression in human endometrial stromal cells and in the pregnant uterus in mice [84]. Apparently, cardiac and uterine *CORIN* expression is controlled by different sets of transcription factors. In the uterus, the progesterone receptor, GATA-2, and KLF17 appear critical for *CORIN* expression. It remains unknown if KLF17 also controls the *NPPA* and *NPR1* expression in the uterus.

## 4. Genetic Loci Associated with Gestational Hypertension and Preeclampsia in Humans

Genetic variants are associated with the risk of gestational hypertension and preeclampsia. In a large-scale sibling pair study, for example, maternal and fetal genetic factors were found to be responsible for >50% of the liability of preeclampsia [85]. Genome-wide association studies (GWAS) have implicated multiple loci, e.g., *PLEKHG1, INHBB*, *FLT1*, *ZNF831*, *FTO, MECOM, FGF5,* and *SH2B3,* in preeclamptic women [86,87,88,89]. The variants in those loci may influence the susceptibility of preeclampsia via the maternal and/or fetal genome, although the underlying molecular mechanisms remain to be elucidated.

Paternal and maternal family history of hypertension is a known risk factor in preeclampsia [90,91]. Consistent with the natriuretic peptide function in regulating blood volume and pressure, variants at the *MTHFR-CLCN6* locus on 1p36, where *NPPA* and *NPPB* genes are present, have been associated with blood pressure levels in human populations [54,92]. In two most recent large-scale GWAS analyses, *MTHFR-CLCN6*, *NPR3*, also called *NPRC* (encoding natriuretic peptide receptor C on 5p13), *FURIN* (15q26), and *PGR* (encoding the progesterone receptor on 11q22) were identified among 18 independent loci associated with gestational hypertension and preeclampsia [18,19]. These findings indicate that genetic variants associated with natriuretic peptide and progesterone signaling may contribute to the development of gestational hypertension and preeclampsia in humans.

As discussed below, ANP plays a critical role in regulating cellular events in endometrial decidualization and trophoblast invasion in the pregnant uterus. Natriuretic peptide receptor C-mediated clearance is one of the major mechanisms in reducing natriuretic peptide levels in vivo [46,48]. In mice, *Nprc* deficiency in adipose tissue enhances natriuretic peptide signaling, which protects against obesity induced by a high-fat diet and adipose tissue inflammation [93,94]. In preeclamptic women, increased NPRC expression has been found in subcutaneous vascular endothelial cells, suggesting a potential role of increased ANP clearance in preeclampsia [95]. Furin is a proprotein convertase that processes many proteins in diverse tissues [96,97]. It is also responsible for pro-BNP and pro-CNP processing [40,43,44]. Progesterone is the main hormone for stimulating endometrial decidualization [69]. Impaired progesterone receptor B signaling has been identified as an underlying mechanism in defective decidualization associated with preeclampsia [98]. The findings from the latest GWAS in humans indicate important functions of natriuretic peptides and progesterone in endometrial decidualization and spiral artery remodeling during pregnancy. More studies are expected to define the clinical value of these genetic variants in gestational hypertension and preeclampsia.

## 5. Gestational Hypertension in ANP- and Corin-Deficient Mice

Despite major differences in uterine and placental structures between humans and rodents [7,79,80], the mouse remains a common model of gestational hypertension and preeclampsia due to its small size, short gestational period, and available techniques to disrupt genes in embryonic stem cells [99]. Consistent with findings in human GWAS analyses, increasing evidence in mouse models supports the idea that defects in ANP signaling may contribute to gestational hypertension and preeclampsia.

### 5.1. ANP KO Mice

ANP is essential for normal blood pressure. In mice, disruption of the *Nppa* gene causes salt-sensitive hypertension [100]. The hypertension is exacerbated when the mice become pregnant, particularly at the late stage of gestation [16,101]. The mice also develop late gestational proteinuria [16], which is common in preeclamptic women [102]. In histological analysis, impaired uterine spiral artery remodeling and trophoblast invasion were evident at the maternal–fetal interface in ANP KO mice, when examined at ~12.5 gestational days [16]. These findings suggest that ANP may promote spiral artery remodeling and trophoblast invasion during pregnancy.

In humans, gestational hypertension is associated with an increased long-term risk of cardiovascular diseases, including cardiac hypertrophy, myocardial infarction, heart failure, and stroke [103,104]. In ANP KO mice, cardiac hypertrophy is exacerbated during pregnancy [105]. Interestingly, ANP^+/−^ offspring from WT males (normotensive) and ANP KO females (with gestational hypertension) are susceptible to developing cardiac hypertrophy and diastolic dysfunction, compared to ANP^+/−^ offspring from ANP KO males (hypertensive) and WT females (no gestational hypertension) [101,106]. Moreover, the ANP^+/−^ offspring from ANP KO females with gestational hypertension also have bigger cerebral infarct volumes following acute ischemic stroke [107]. In these mice, altered gene profiles are found in major organs, including the heart, kidney, and brain [101,106,107]. These findings underscore the importance of ANP in uterine biology and the long-term genetic impact of ANP deficiency on cardiovascular fitness.

### 5.2. Corin KO Mice

Corin is necessary for ANP activation [38]. Consistent with findings in the pregnant ANP KO mice, gestational hypertension and proteinuria occur in pregnant *Corin* KO mice, starting at ~17 gestational days [16]. There are delayed spiral artery remodeling and trophoblast invasion at the maternal–fetal interface, renal ischemic lesions, and placental necrosis and calcification in the pregnant *Corin* KO mice [16]. The pathological phenotype persisted when cardiac, but not uterine, corin expression was restored by a heart-specific *Corin* transgene in the KO mice [16]. The results indicate that the lack of uterine, but not cardiac, corin is likely responsible for the preeclampsia-like phenotype in *Corin* KO mice.

In addition to the heart, corin is expressed in the placenta [108,109]. In principle, the phenotype in *Corin* KO mice could result from deficiencies in maternal or fetal corin or both. To distinguish the role of maternal from fetal corin, female *Corin* KO mice were mated with WT or *Corin* KO males, resulting in embryos with one or no copy of the functional *Corin* allele. Gestational hypertension was similar in *Corin* KO females mated with WT or *Corin* KO males [16], indicating that the lack of maternal, but not fetal, corin contributed to gestational hypertension. As in ANP KO mice, pregnant *Corin* KO mice developed cardiac hypertrophy, which continued postpartum and worsened with age [110]. In humans, gestational hypertension and peripartum cardiomyopathy are more likely to occur in older pregnant women [111]. These findings indicate that corin deficiency may be a contributing factor in pregnancy-associated hypertension and heart disease.

A serine protease of the chymotrypsin-fold, corin is synthesized as zymogen, which is activated by proprotein convertase subtilisin/kesin 6 (PCSK6) [112]. In mice, *Pcsk6* deficiency prevents corin activation, leading to salt-sensitive hypertension [112]. In a rat model of pregnancy-induced hypertension associated with hyperinsulinemia, reduced corin and PCSK6 expression was found in the placenta [108]. On the other hand, increased PCSK6 expression was reported in placental samples from preeclamptic patients [113]. These findings suggest a potential role of aberrant PCSK6 expression in the pathogenesis of preeclampsia. More studies are needed to determine if PCSK6 cleaves other substrates in the placenta and if altered PCSK6 expression contributes to gestational hypertension and preeclampsia in humans.

### 5.3. Klf17 KO Mice

KLF17 is a transcription factor of the KLF family [114]. In humans, KLF17 inhibits epithelial–mesenchymal transition and cancer progression via TGFβ/Smad/p53 signaling [115,116]. *KLF17* downregulation has been found in many cancers [115,117]. In *Xenopus* and zebrafish, Klf17 is essential for embryogenesis [118,119]. In mice, *Klf17* expression is upregulated in the pregnant uterus [84]. Increased uterine *Klf17* expression was also found in ovariectomized mice treated with progesterone, consistent with its upregulation in pregnancy [84]. As discussed above, disruption or inhibition of the *KLF17* gene prevented *CORIN* expression in human endometrial stromal cells [84], suggesting that KLF17 may be a mediator in uterine corin expression and function.

Both male and female *Klf17* KO mice are viable and fertile [84]. In pregnant *Klf17* KO mice, uterine corin expression was undetectable, in agreement with the cell-based promotor studies [84]. In contrast, cardiac corin expression was not altered in *Klf17* KO mice. As in ANP and *Corin* KO mice, a preeclampsia-like phenotype was observed in *Klf17* KO mice, including late gestational hypertension, renal ischemia, and proteinuria [84]. *Klf17* KO mice also had impaired spiral artery remodeling, delayed trophoblast invasion, and smaller litter sizes, when examined at ~12.5 gestational days [84]. These findings show that Klf17 deficiency prevents uterine corin expression, resulting a preeclampsia-like phenotype in mice. It will be important to verify these findings in humans. Given the fact that KLF17 is a transcription factor, KLF17 likely regulates other genes in the uterus. More investigations are required to test this hypothesis. The results may provide new insights into the molecular mechanisms underlying uterine tissue remodeling in pregnancy.

## 6. ANP in Uterine and Placental Cell Phenotype and Function

### 6.1. The Soil, Seeds, and Roots

Endometrial decidualization involves a cascade of gene expression and cellular changes necessary for embryo implantation and trophoblast invasion [69]. Trophoblast invasion is a process, in which trophoblast cells from the developing embryo invade the decidualized endometrium and make connections with maternal spiral arteries [120,121]. This process is crucial for the access of an adequate maternal blood supply to nourish the developing fetus.

In an analogy of soil and seed, the decidualized endometrium represents the receptive soil that provides a fertile environment for invading trophoblasts, i.e., the seeds. For the trophoblasts to grow effectively, endometrial decidualization must be adequate. Upon entering the decidualized endometrium, the trophoblasts interact with the decidual cells, extend invasive projections, like roots, into the decidualized tissue, and release signaling molecules to modify the endometrium, making it easier for the roots to extend. Such an intricate interplay between the decidual cells and the invading trophoblasts is critical for proper spiral artery remodeling, a healthy maternal–fetal interface, and an ultimately successful pregnancy.

Poor spiral artery remodeling, reduced uteroplacental blood flow, and ensuing placental ischemia have been recognized as central pathological mechanisms in preeclampsia. Increased systemic blood pressure in preeclamptic women may reflect, in part, a compensatory response to placental ischemic signals, to maintain steady blood flow in the presence of narrow uterine spiral arteries. Endometrial decidualization is essential for spiral artery remodeling [69,122]. Thus, poorly decidualized endometrium could be a major reason for defective spiral artery remodeling in preeclampsia. Indeed, studies with uterine samples from preeclamptic women have shown altered gene and protein profiles that are indicative of improper endometrial decidualization and sFlt1 expression [10,123]. Decreased circulating levels of decidualization markers, such as prolactin and IGFBP-1, have also been found in preeclamptic women [11,124]. These findings support the idea that defective endometrial decidualization is an important mechanism in the pathogenesis of preeclampsia. These studies also point to ANP signaling as a potential regulatory mechanism in early cellular events of decidualization and spiral artery remodeling, as well as in subsequent trophoblast invasion in the pregnant uterus.

### 6.2. Role of ANP in Decidual and Vascular Cell Interactions

#### 6.2.1. Decidualization and Spiral Artery Remodeling

Endometrial decidualization and uterine spiral artery remodeling occur before direct contact with invading trophoblasts [70]. In fact, morphological changes in endometrial arteries were observed in women with ectopic pregnancies, i.e., the implantation occurs outside the uterus [70]. These findings indicate that the initial cellular events in spiral artery remodeling are mediated mostly by signaling molecules from decidual stromal cells, but not trophoblasts.

Studies in animal models also support the role of decidual stromal cells in regulating cellular changes in spiral arteries. In mice, for example, apoptosis is markedly increased in the pregnant uterus [15]. Sequential death in smooth muscle cells and endothelial cells was observed in endometrial arterial segments, where no adjacent intravascular or interstitial trophoblast invasion was detected [15]. In cell culture, decidualized human endometrial stromal cells were found to secrete TNF-related apoptosis-inducing ligand (TRAIL), a death protein that triggered apoptosis in uterine smooth muscle cells [15]. Subsequently, the apoptotic smooth muscle cells released cyclophilin B, a proinflammatory protein that upregulates TRAIL receptor in cancer cells via MAPK/ERK signaling [125,126]. Similar cyclophilin B-mediated MAPK/ERK signaling and TRAIL receptor upregulation were observed in cultured uterine endothelial cells. As a result, the uterine endothelial cells became responsive to TRAIL-induced cell death [15]. These findings could explain the observed sequential cell death, first in arterial smooth muscle cells and then in endothelial cells in the mouse pregnant uterus.

In pregnant ANP KO mice, uterine endometrial decidualization is compromised, as indicated by reduced prolactin and IGFBP-1 expression [15]. When cultured human endometrial stromal cells were treated with ANP, prolactin, IGFBP-1, and TRAIL expression was increased. The conditioned medium from the ANP-treated endometrial stromal cells also had an increased activity in inducing apoptosis in uterine smooth muscle cells [15]. On the other hand, ANP treatment in cultured human endothelial cells did not alter cell proliferation and migration or prevent TNFα-induced dysfunction [127]. These findings are consistent, indicating an important role for ANP in inducing TRAIL expression and release from endometrial stromal cells, causing apoptosis in arterial smooth muscle cells [15]. Cyclophilin B release from smooth muscle cells and TRAIL receptor induction eventually lead to apoptosis in arterial endothelial cells. These well-orchestrated cellular events are probably an important part of the spiral artery remodeling process. The loss of arterial smooth muscle cells and endothelial cells may pave the way for intravascular invasion of placental trophoblasts. More studies are required to understand the ANP-mediated signaling mechanism in uterine stromal cells.

ANP is expressed in the uterus and placenta [17,74,128]. In principle, the effects on decidual stromal cells could be due to ANP from either the uterus or the placenta, or both. To distinguish the role of uterine ANP from placental ANP, mouse models with either uterine or placental ANP deficiency were analyzed [15]. As indicated by reduced prolactin and IGFBP1 expression, uterine decidualization was defective in mice lacking either uterine or placental ANP, indicating that ANP from both tissues contributes to the decidualization process [15]. However, impaired decidualization was more profound in mice lacking uterine ANP than placental ANP, indicating that ANP made in the uterus is most critical for decidualization. Consistently, studies in *Corin* KO mice also show that uterine corin is more important than placental corin in promoting spiral artery remodeling [16]. In addition, low uterine TRAIL levels were observed in pregnant ANP and *Corin* KO mice [15], which exhibit a preeclampsia-like phenotype. Thus, in addition to its vasorelaxation function, uterine ANP, activated by corin in situ, may promote decidualization and TRAIL expression in endometrial stromal cells. The stromal cell-derived TRAIL induces arterial smooth muscle cell death and cyclophilin B release. Cyclophilin B in turn upregulates TRAIL receptor expression in endothelial cells, subjecting the cells to TRAIL-induced apoptosis (Figure 3). Defects in the corin and ANP function are expected to impair endometrial decidualization and spiral artery remodeling, thereby contributing to gestational hypertension and preeclampsia. Additional investigations will be important to verify this hypothesis in other animal models and to test if a similar mechanism exists in the human pregnant uterus. It will be also important to examine if ANP signaling contributes to uterine structural changes during the normal menstrual cycle in humans.

#### 6.2.2. Trophoblast Invasion

As discussed above, the initiation of endometrial decidualization and spiral artery remodeling is independent of trophoblasts [69]. However, embryo implantation and the subsequent trophoblast invasion are expected to act in a reciprocal manner to enhance endometrial decidualization (Figure 2). It is well documented that trophoblasts can secrete a variety of proteins, such as proteolytic enzymes, cytokines, and signaling molecules, to promote endometrial decidualization, extracellular matrix degradation, and uterine tissue remodeling [129,130,131]. Trophoblasts also secrete proapoptotic factors, such as TRAIL, TNFα, and FAS ligand, to induce apoptosis in arterial smooth muscle cells and endothelial cells, as shown in cell experiments and mouse models [132]. These activities are expected to accelerate the structural changes in arterial walls and facilitate trophoblast penetration, leading to the transformation of the uterine arteries from high- to low-resistance vessels. Such transformation ensures an increased uteroplacental blood flow with low velocity, which is necessary for supporting the growth of the developing fetus [133].

The placenta is another site for corin and pro-ANP expression, indicating a role of ANP in trophoblast function [15,108,109]. In supporting this hypothesis, the ANP receptor NPR-A is expressed in human trophoblasts and decidual stromal cells [16,71]. In cultured human trophoblasts, ANP treatment increases intracellular cGMP levels and promotes trophoblast invasion in Matrigel assays, whereas NPR-A knockdown suppresses trophoblast invasion [16,73,134]. Moreover, markedly decreased placental corin and NPR-A expression and increased levels of pro-ANP are found in preeclamptic women [73,134]. These results indicate that ANP-mediated signaling probably has a role in promoting trophoblast invasion, and that defects in ANP signaling may contribute to uterine and/or placental pathology in preeclampsia.

The ANP signaling mechanism in trophoblasts is not fully understood. In the heart, ANP modulates autophagy, a process of regulating cellular function depending on pathophysiological conditions [135,136]. In a recent study in human trophoblasts and chorionic villi, a molecular mechanism has been suggested, in which ANP activates NPR-A and subsequent adenosine 5′-monophosphate-activated protein kinase (AMPK) and the mammalian target of rapamycin complex 1 (mTORC1) signaling to inhibit autophagy and induce protein kinase N3 (PKN3) expression, thereby increasing metalloproteinase expression and trophoblast invasiveness [134] (Figure 4). Autophagy activation or downregulation of PKN3 diminishes the effect of ANP on trophoblast invasion [134]. Moreover, placentas from preeclamptic women had reduced PKN3 protein levels [134]. These results are intriguing, indicating a potential link between impaired ANP signaling and low PKN3 activity in the pathogenesis of preeclampsia. It appears that ANP can stimulate multiple signaling pathways to modify a wide range of cellular functions in endometrial decidualization and trophoblast invasion, which are important in pregnancy.

## 7. Conclusions and Perspective

Preeclampsia is a major health problem in pregnant women. Defective uterine spiral artery remodeling plays a key role in the pathogenesis of preeclampsia. Adequate remodeling of spiral arteries depends on systemic and local hormonal actions to control complex cellular interactions among decidual cells, endometrial vascular cells, and placental trophoblasts. Defective endometrial decidualization, which prevents spiral artery remodeling and trophoblast invasion, has been recognized as another important mechanism in preeclampsia.

ANP is essential for sodium homeostasis and normal blood pressure. ANP also acts in the pregnant uterus to promote spiral artery remodeling and trophoblast invasion. In large-scale GWAS analyses, genetic loci near natriuretic peptide-related genes are associated with gestational hypertension and preeclampsia in humans [18,19]. Recent molecular and cellular studies have shown that uterine ANP enhances endometrial decidualization, leading to morphological changes in vascular cells via a TRAIL-dependent mechanism [15]. In trophoblasts, ANP signaling inhibits autophagy and increases PKN3 and metalloproteinase expression, promoting trophoblast invasion [134]. These findings highlight the importance of natriuretic peptide signaling in uterine biology and maternal health, suggesting that defects in ANP and related molecules may be an underlying mechanism in the pathogenesis of gestational hypertension and preeclampsia.

In failing hearts, reduced natriuretic peptide activity is common [137,138]. Strategies to increase natriuretic peptide activity have been used to treat patients with heart failure [139,140,141,142]. The recent findings regarding natriuretic peptide signaling in uterine biology and function should encourage more investigations to verify these results and to test if similar strategies to increase uterine ANP production or activity may be used to treat preeclampsia.

## Figures and Tables

**Figure 1 ijms-24-12309-f001:**
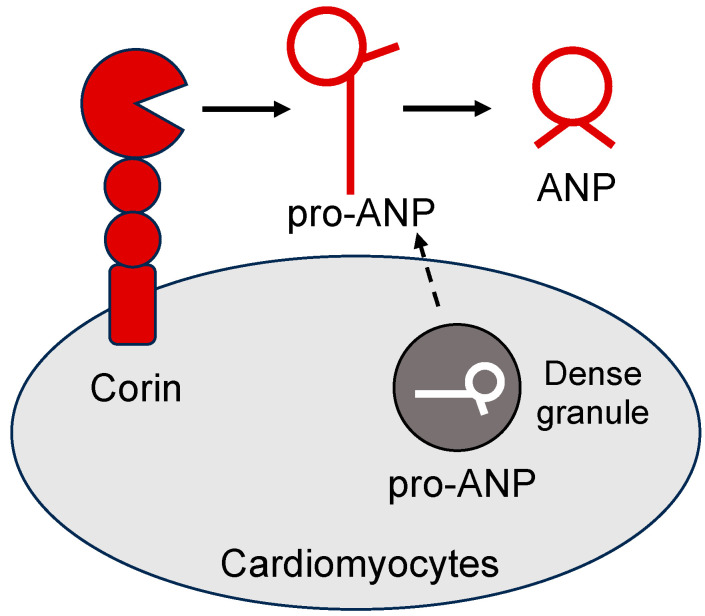
Corin-mediated pro-ANP processing in cardiomyocytes. Corin is a type II transmembrane serine protease anchoring on the cell surface. In cardiomyocytes, pro-ANP is stored in the dense granules. Upon stimulation, pro-ANP is secreted from cardiomyocytes and cleaved by corin, generating biologically active ANP.

**Figure 2 ijms-24-12309-f002:**
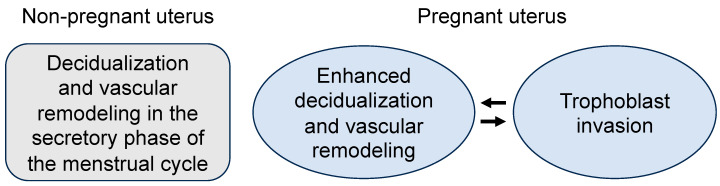
Illustration of endometrial decidualization, vascular remodeling, and trophoblast invasion in non-pregnant and pregnant uteruses. In the non-pregnant uterus, endometrial decidualization and vascular remodeling occur in the secretory phase of the menstrual cycle after ovulation. In the pregnant uterus, endometrial decidualization and vascular remodeling are accelerated upon embryo implantation, which facilitates trophoblast invasion. The invading trophoblasts in turn stimulate endometrial decidualization and vascular remodeling in a reciprocal manner.

**Figure 3 ijms-24-12309-f003:**
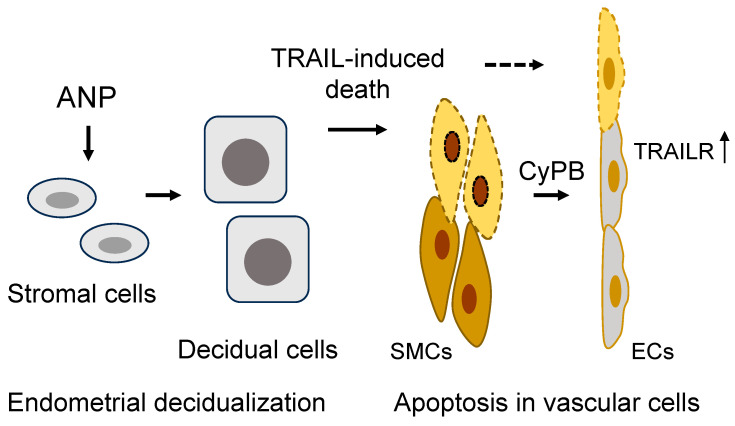
A model of ANP function in endometrial decidualization and spiral artery remodeling in the pregnant uterus. In the pregnant uterus, ANP stimulates stromal cell decidualization and TNF-related apoptosis-inducing ligand (TRAIL) expression and secretion, inducing apoptosis in arterial smooth muscle cells (SMCs). Apoptotic SMCs release cyclophilin B (CyPB) to upregulate TRAIL receptor (TRAILR) expression in endothelial cells (ECs), making the cells susceptible to TRAIL-mediated apoptosis. The sequential death in SMCs and ECs facilitates spiral artery remodeling.

**Figure 4 ijms-24-12309-f004:**
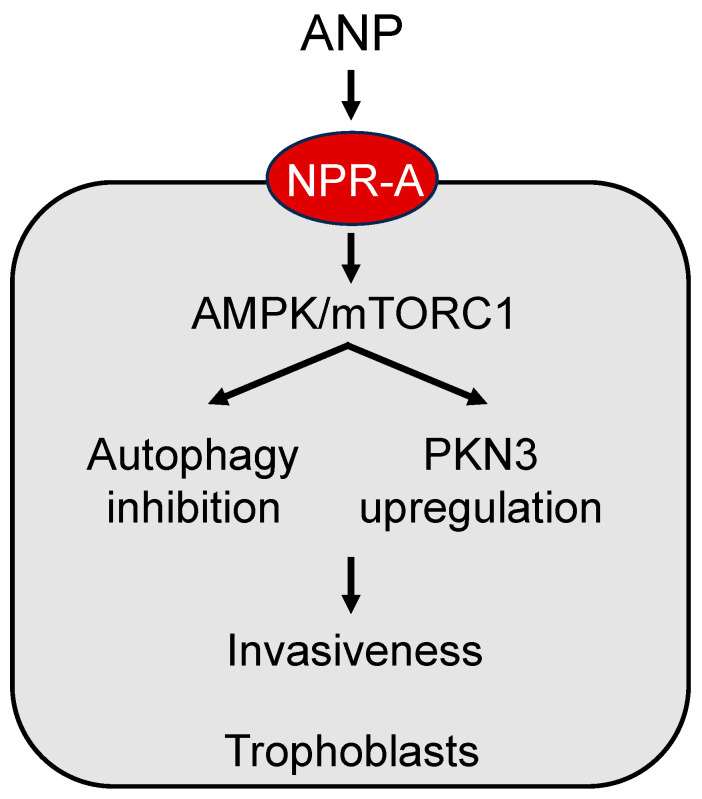
A model of ANP signaling in placental trophoblasts. ANP activates natriuretic peptide receptor A (NPR-A) to enhance adenosine 5′-monophosphate-activated protein kinase (AMPK) and the mammalian target of rapamycin complex 1 (mTORC1) signaling, leading to autophagy inhibition and protein kinase N3 (PKN3) upregulation, thereby increasing trophoblast invasiveness.

## Data Availability

Not applicable.

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
