# Peer review of "Natriuretic Peptide Signaling in Uterine Biology and Preeclampsia"

_ijms, 2023, doi:10.3390/ijms241512309_

Round 1

Reviewer 1 Report

In this manuscript, Wu provides an overview of natriuretic peptide and its role in uterine biology and preeclampsia.  The background on ANP in Section 2 discussing synthesis, signaling and known functions provides key background for discussion of its importance in decidualization and pregnancy/preeclampsia. Section 3 is a bit confusing because the title implies it is about the pregnant uterus, but most of the information involves events prior to implantation.  As they point out in lines 316-317 (Section 5), in humans implantation is not a prerequisite for endometrial decidualization. Providing this information earlier in the review would be useful and potentially help put Section 3 in better context.  In addition, pointing out differences between mouse/rat pregnancy/decidualization and human events is important for interpretation of the studies.  Another figure detailing the events of decidualization in relation to the normal menstrual cycle with implantation leading to early pregnancy would be helpful.  Organizing sections based on what ANP does prior to implantation in the non-pregnant uterus vs after implantation might be useful, assuming the term pregnancy means once implantation occurs.  Also some organization detailing what ANP does in a normal pregnancy and what changes in a preeclamptic pregnancy would be good. This would also be helpful in putting Fig 2 into context i.e. the time course of changes in vascular cells in relation to decidualization and implantation. The current Figures are definitely helpful in illustrating points, but more illustration would strengthen the manuscript.  Section 6 provides a strong rationale for looking at ANP given the human genetics data.  Moving this section forward might have more impact.  The best final re-organization depends on the transitions and the major points to be made in the context of the timeline of events in the initial decidualization, implantation, trophoblast invasion, endothelial cell changes, spiral artery remodeling, placental development in normal vs situations with adverse pregnancy outcomes.

Minor concerns

1.  Sentence structure, spelling, or improper verbs or prepositions in some places.

Line 34, 151-152, 181-182, 233, 329, 371-373, 449-450,

Paragraph in lines 271-282 is italicized for unknown reasons.

1.  Sentence structure, spelling, or improper verbs or prepositions in some places.

Line 34, 151-152, 181-182, 233, 329, 371-373, 449-450,

Author Response

In this manuscript, Wu provides an overview of natriuretic peptide and its role in uterine biology and preeclampsia. The background on ANP in Section 2 discussing synthesis, signaling and known functions provides key background for discussion of its importance in decidualization and pregnancy/preeclampsia. Section 3 is a bit confusing because the title implies it is about the pregnant uterus, but most of the information involves events prior to implantation.  As they point out in lines 316-317 (Section 5), in humans implantation is not a prerequisite for endometrial decidualization. Providing this information earlier in the review would be useful and potentially help put Section 3 in better context.

Response: The reviewer’s point is appreciated. To avoid the confusion, the title of Section 3 has been revised to “Expression of ANP and Corin in the Uterus” (line 183). In addition, a new paragraph has been added to the beginning of this section to indicate that in humans, embryo implantation is not a prerequisite for endometrial decidualization (lines 184-196).

In addition, pointing out differences between mouse/rat pregnancy/decidualization and human events is important for interpretation of the studies. Another figure detailing the events of decidualization in relation to the normal menstrual cycle with implantation leading to early pregnancy would be helpful. Organizing sections based on what ANP does prior to implantation in the non-pregnant uterus vs after implantation might be useful, assuming the term pregnancy means once implantation occurs.

Response: We thank the reviewer for the suggestion. Accordingly, the text has been revised to point out that endometrial decidualization, uterine artery remodeling, and trophoblast invasion are less extensive in many mammalian species, particularly in rodents, compared to those in humans (lines 230-232 and 285-288). A new Figure 2 has been added to illustrate events of endometrial decidualization, spiral artery remodeling, and trophoblast invasion in non-pregnant and pregnant uteruses. Currently, the potential function of ANP signaling in the non-pregnant uterus remains unknown. The text has been revised to point out that “It will be also important to examine if ANP signaling contributes to uterine structural changes during the normal menstrual cycle in humans” (lines 454-455).

Also some organization detailing what ANP does in a normal pregnancy and what changes in a preeclamptic pregnancy would be good. This would also be helpful in putting Fig 2 into context i.e. the time course of changes in vascular cells in relation to decidualization and implantation. The current Figures are definitely helpful in illustrating points, but more illustration would strengthen the manuscript.

Response: As indicated above, a new Figure 2 has been added to illustrate endometrial decidualization, vascular remodeling, and trophoblast invasion in non-pregnant and pregnant uteruses. The text has been revised accordingly.

Section 6 provides a strong rationale for looking at ANP given the human genetics data.  Moving this section forward might have more impact.

Response: This is a good suggestion. Accordingly, this section has been moved forward (new Section 4).

The best final re-organization depends on the transitions and the major points to be made in the context of the timeline of events in the initial decidualization, implantation, trophoblast invasion, endothelial cell changes, spiral artery remodeling, placental development in normal vs situations with adverse pregnancy outcomes.

Response: The text, including the Abstract and Sections 3-7, has been revised based on the revised organization and transitions.

Minor concerns

  1. Sentence structure, spelling, or improper verbs or prepositions in some places.

Line 34, 151-152, 181-182, 233, 329, 371-373, 449-450.

Response: Typos and grammatical errors listed above have been fixed.

Paragraph in lines 271-282 is italicized for unknown reasons.

Response: This was made during editorial editing. It is fixed now.

Reviewer 2 Report

The review entitled: "Natriuretic Peptide Signaling in Uterine Biology and  Preeclampsia" by Qingyu Wu is an excellent paper which summarizes the involvement of the natriuretic peptide system in the development of normal pregnancy and preeclampsia.

The review has neatly outlined the subtitles and elegantly summarizes all the relevant literature.

I have only few minor issues that should be addressed before publications:

1- The author did not refer to VEGF and sFlt as major player of preeclampsia. 

I am aware of the focus of this review, namely NPs family and preeclampsia, however, brief referral to VEGF/sFlt1 should be added.

Actually, the author cited one relevant MS in this context: Sahu MB, Deepak V, Gonzales SK, Rimawi B, Watkins KK, Smith AK, et al. Decidual cells from women with preeclampsia 683 exhibit inadequate decidualization and reduced sFlt1 suppression. Pregnancy Hypertens. 2019;15:64-71.

2- The author devoted great attention to Corin which is the right thing to do, however i expect to refer also to the enzyme PCSK6 that activates corin. It may be involved in the pathogenesis of preeclampsia. 

Author Response

The review entitled: "Natriuretic Peptide Signaling in Uterine Biology and Preeclampsia" by Qingyu Wu is an excellent paper which summarizes the involvement of the natriuretic peptide system in the development of normal pregnancy and preeclampsia.

The review has neatly outlined the subtitles and elegantly summarizes all the relevant literature.

Response: The reviewer’s highly positive comments are greatly appreciated.

I have only few minor issues that should be addressed before publications:

1- The author did not refer to VEGF and sFlt as major player of preeclampsia.

I am aware of the focus of this review, namely NPs family and preeclampsia, however, brief referral to VEGF/sFlt1 should be added.

Actually, the author cited one relevant MS in this context: Sahu MB, Deepak V, Gonzales SK, Rimawi B, Watkins KK, Smith AK, et al. Decidual cells from women with preeclampsia exhibit inadequate decidualization and reduced sFlt1 suppression. Pregnancy Hypertens. 2019;15:64-71.

Response: Indeed, this review is primarily on natriuretic peptide signaling in uterine biology and preeclampsia. The reviewer’s point is well taken. The Introduction has been revised to indicate that aberrant expression and/or function of VEGF and its receptors, e.g., sFlt1, play a major role in the pathogenesis of preeclampsia (lines 41-45 and 390).

2- The author devoted great attention to Corin which is the right thing to do, however i expect to refer also to the enzyme PCSK6 that activates corin. It may be involved in the pathogenesis of preeclampsia.

Response: This is a good point. Accordingly, a new paragraph has been added to point out that PCSK6 is the corin activator and may be involved in the pathogenesis of gestational hypertension and preeclampsia (lines 332-341).

Round 2

Reviewer 1 Report

The authors were quite responsive to the critique and have greatly improved the manuscript with a re-arrangement as well as the addition of a figure and clarifying and transitional paragraphs.  A few more minor concerns should be considered, along with more and continued editing of the English in the manuscript.

Minor concern:

Technically, preeclampsia should not be referred to as a disease (lines 10 and 34) throughout the manuscript, but as a syndrome or disorder of pregnancy (https://pubmed.ncbi.nlm.nih.gov/32679789/). 

Line 209

“Gene expression is a more specific way to analyze ANP expression in non-cardiac tissues.” Should probably more precisely read:  Gene expression is a more specific way to analyze the potential for expression of ANP protein in non-cardiac tissues.

Line 250

“Genetic variants alter the risk of gestational hypertension and preeclampsia” should more precisely read “Genetic variants are associated with the risk of gestational hypertension and preeclampsia”

English needs further editing, including the following lines. 

Line 35, 285, 304, 307, 315,

However, this list may not be all inclusive and the English needs to be carefully checked throughout the revised manuscript.

See comments to authors

Author Response

The authors were quite responsive to the critique and have greatly improved the manuscript with a re-arrangement as well as the addition of a figure and clarifying and transitional paragraphs.  A few more minor concerns should be considered, along with more and continued editing of the English in the manuscript.

Response: The reviewer’s positive comments are appreciated.

Minor concern:

Technically, preeclampsia should not be referred to as a disease (lines 10 and 34) throughout the manuscript, but as a syndrome or disorder of pregnancy (https://pubmed.ncbi.nlm.nih.gov/32679789/).

Response: The word of ‘disorder’ is used now (lines 10 and 34).

Line 209

“Gene expression is a more specific way to analyze ANP expression in non-cardiac tissues.” Should probably more precisely read:  Gene expression is a more specific way to analyze the potential for expression of ANP protein in non-cardiac tissues.

Response: Thank you for the suggestion. The sentence has been revised (lines 209-210).

Line 250

“Genetic variants alter the risk of gestational hypertension and preeclampsia” should more precisely read “Genetic variants are associated with the risk of gestational hypertension and preeclampsia.”

Response: The sentence has been revised, as suggested (lines 250-251).

English needs further editing, including the following lines.

Line 35, 285, 304, 307, 315.

Response: The sentences indicated above have been revised (lines 35, 281-283, 299-300, 303-304, and 310-312).

However, this list may not be all inclusive and the English needs to be carefully checked throughout the revised manuscript.

Response: Thank you for pointing this out. Accordingly, additional revisions have been made throughout the manuscript (highlighted in red) to improve the English writing.